Journal of Machine Learning Research (2024) 1-11    Submitted COMPAYL 2024; Published

# Preprocessing Pathology Reports for Vision-Language Model Development

**Ruben T. Lucassen**                                                R.T.LUCASSEN@UMCUTRECHT.NL
*Dept. of Pathology, University Medical Center Utrecht, the Netherlands*
*Dept. of Biomedical Engineering, Eindhoven University of Technology, the Netherlands*

**Tijn van de Luijtgaarden**
*Dept. of Mathematics and Computer Science, Eindhoven University of Technology, the Netherlands*

**Sander P. J. Moonemans**
*Dept. of Mathematics and Computer Science, Eindhoven University of Technology, the Netherlands*

**Willeke A. M. Blokx**
*Dept. of Pathology, University Medical Center Utrecht, the Netherlands*

**Mitko Veta**
*Dept. of Biomedical Engineering, Eindhoven University of Technology, the Netherlands*

**Editor:**

## Abstract

Pathology reports are increasingly being used for development of vision-language models. Because the reports often include information that cannot directly be derived from paired images, careful selection of information is required to prevent hallucinations in tasks like report generation. In this paper, we present a language model for subsentence segmentation based on the information content, as part of a preprocessing workflow for 27,500 pathology reports of cutaneous melanocytic lesions. After initial clean up, the reports were first translated from Dutch to English and then segmented by separate language models. Both models were developed using an iterative approach, in which the development dataset was expanded with manually corrected model predictions for previously unannotated reports before finetuning the next version of the models. Over the course of eight iterations, the development dataset was in the end scaled up to 1,500 translated and annotated reports. On the independent test set of 3,597 sentences from 150 reports, 219 translation errors (6,1%) of different severities were counted. The subsentence segmentation model achieved a strong predictive performance on the test set with a macro average $F_1$-score of 0.921 (95% CI, 0.890-0.940) and a weighted average $F_1$-score of 0.952 (95% CI, 0.944-0.960) over 13 different classes. The remaining 25,850 unannotated reports were translated and segmented using the final models to complete the dataset preprocessing. Differences in word count and class distribution between section types of the reports were explored in preparation for future vision-language modeling. The presented methodology is generic and can, therefore, easily be extended to multiple or different pathology domains beyond melanocytic skin lesions. Code and trained model parameters are made publicly available.

**Keywords:** pathology report, language models, translation, subsentence segmentation

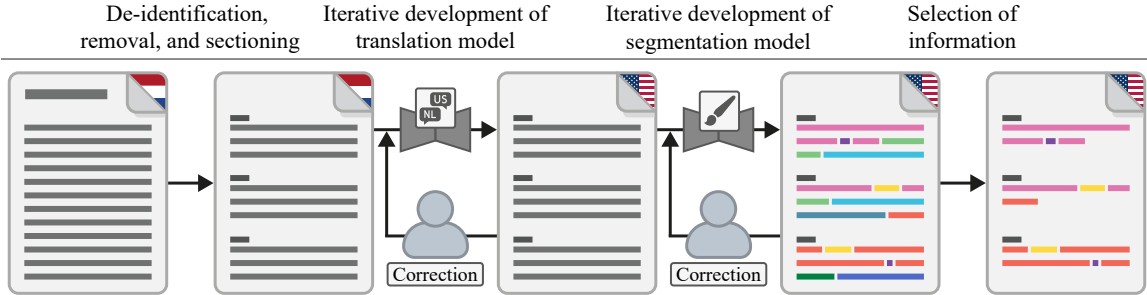

Figure 1: Overview of the pathology report preprocessing workflow.

## 1 Introduction

Vision-language models in computational pathology have recently gained attention for tasks like multimodal pre-training, image retrieval, and image captioning. (Huang et al., 2023; Lu et al., 2024b; Vorontsov et al., 2024; Lu et al., 2024a) Some of the first models that combined vision and language for histopathology, such as PLIP (Huang et al., 2023) and CONCH (Lu et al., 2024b), were trained using histopathology images paired with captions, primarily scraped from social media or scientific literature. However, for case-level tasks like pathology report generation, which can potentially reduce the workload of pathologists, digitally archived whole slide images and corresponding pathology reports likely form a more suitable resource for vision-language model development. This is in line with more recent models, such as PRISM (Vorontsov et al., 2024) and PathChat (Lu et al., 2024a).

Pathology reports are often diverse in content, including sentences about cell and tissue appearance on hematoxylin and eosin (H&E)-stained slides, expression patterns in immuno-histochemistry (IHC), results of molecular testing, clinical information, patient history, (differential) diagnoses, and treatment recommendations. (Scolyer et al., 2013) Using sentences with information that cannot be derived from the paired images during training can lead to hallucinations (i.e., generated statements that contradict the source content) (Ji et al., 2023), which can be dangerous for high-stakes applications such as in healthcare, motivating the need for careful selection of information from the reports. Whereas preprocessing steps for whole slide images (e.g., tessellation, pen marking removal, and stain normalization) are well-documented and extensively studied, pathology report preprocessing for vision-language modeling is comparatively new and best-practices are still developing.

In this work, we present a language model for segmentation of subsentences based on the information content as part of a preprocessing workflow for pathology reports. After de-identification, removal of irrelevant sentences, and sectioning, the reports were first translated from Dutch to English and then segmented by separate language models, both developed using an iterative model development approach (Figure 1). While the results shown in this paper are specific to reports of cutaneous melanocytic lesions, the presented methodology is more general and can straightforwardly be extended to multiple or different pathology domains. The code and trained model parameters are made publicly available[1].

---

1. https://github.com/RTLucassen/report_preprocessing

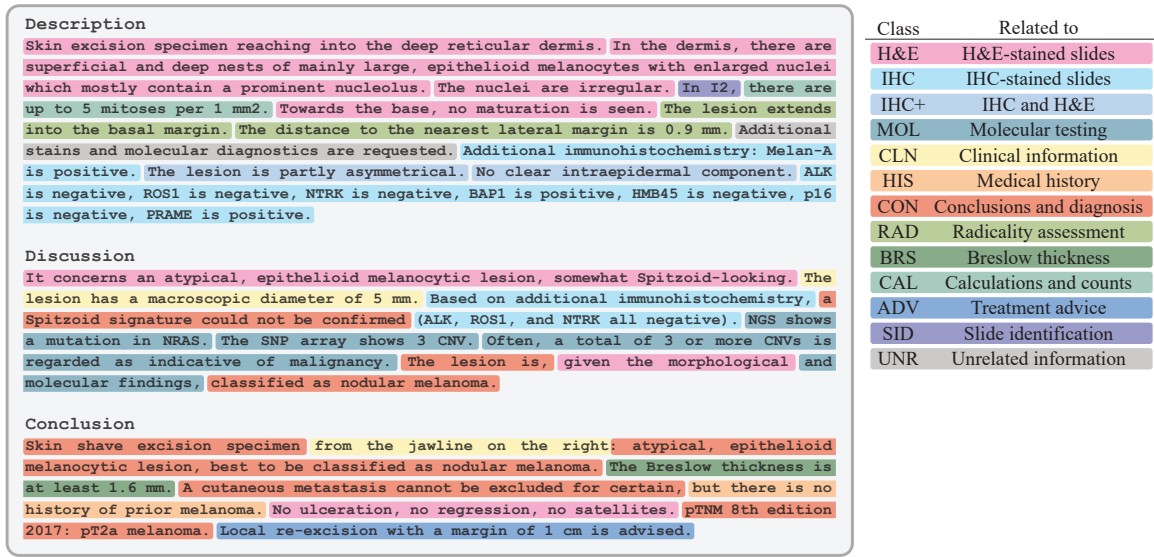

Figure 2: Example of an annotated pathology report describing a cutaneous melanocytic lesion (left) with the corresponding class definitions (right).

## 2 Materials and Methods

### 2.1 Dataset

The dataset used in this work consists of 27,500 pathology reports describing skin biopsy and excision specimens with melanocytic lesions, accessioned between January 1, 2013 and December 31, 2020, and obtained from the digital archive of the Department of Pathology at the University Medical Center Utrecht, the Netherlands. The reports were made as part of routine clinical practice by different general pathologists and more specialized dermatopathologists. The study was conducted in compliance with the hospital's research ethics committee guidelines. Reports for patients who opted out of the use of their data for research purposes were excluded.

Several manual preprocessing steps were performed at the start to clean up the pathology reports. All reports were de-identified by removing names or abbreviations of pathologists and hospitals, as well as dates and case numbers. Identifying patient information was never included in the reports, with the exception of sex and age at accession. Irrelevant sentences were also removed. If multiple unique specimens were originally combined into a single report, the original report was divided into a separate report for every specimen. Each report can include four sections: (1) a structured report; (2) a free-text description of all observations and results of diagnostic testing; (3) a free-text discussion of the findings and the interpretation thereof; (4) a structured or free-text conclusion with key findings, the (differential) diagnosis, and treatment recommendations. Not all sections are present in each report.

Separate language models were used for the translation and segmentation task. Both models were iteratively improved over the course of eight iterations by expanding the development dataset with manually corrected model predictions for previously unannotated

Table 1: Dataset statistics.

| Set | Reports | Sentences | Words |
|---|---|---|---|
| Training | 1,350 | 20,563 | 235,194 |
| Validation | 150 | 2,398 | 27,593 |
| Test | 150 | 3,597 | 37,465 |
| Unannotated | 25,850 | 179,358 | 1,980,246 |
| Total | 27,500 | 205,916 | 2,280,498 |

reports before finetuning the next version of the models, starting each training run from the original model parameters. The number of reports, sentences, and words per subset of the dataset are shown in Table 1. The development set (i.e., training and validation) was in the end scaled up to 1,500 reports, which were in part randomly selected and in part selected to enrich for rare melanocytic lesion subtypes, specimens with a melanocytic lesion in combination with non-melanocytic skin pathologies, as well as reports with previously unseen words for the translation model and uncommon classes for the segmentation model. The number of reports and selection strategy per iteration are shown in Table 3 in the Supplementary Material. The reports were translated and corrected by two translators (R.L., T.L.), annotated and corrected by three annotators (R.L., T.L., S.M.), all under supervision of a pathologist (W.B.). Additionally, a test set of 150 reports, selected similarly from different patients, was annotated independently (i.e., not starting from model predictions) by a single annotator (R.L.) and verified by a pathologist (W.B.) for evaluation of the models. An example of an annotated report with the 13 defined classes is shown in Figure 2. Punctuation marks were included in the annotations in such a way that a sentence remains coherent as much as possible when classes would be filtered out. Annotation and correction of model predictions was performed using the open-source Label Studio software[2]. The remaining 25,850 reports were translated using the final translation model and identified translation errors that remained were corrected before subsentence segmentation by the final segmentation model.

## 2.2 Translation Model

The vast majority of the pathology reports in the dataset were originally written in Dutch (99.7% in Dutch, 0.3% in English). We decided to translate the reports from Dutch to English, as most of the publicly available, pre-trained language models have English as primary language, which can potentially be useful in the downstream development of vision-language models for computational pathology.

We built upon a pre-trained Dutch-to-English translation model, `opus-mt-nl-en`[3], from the OPUS-MT project (Tiedemann and Thottingal, 2020), which is a Transformer model (Vaswani et al., 2017) with an encoder-decoder architecture of 182 million parameters, using SentencePiece (Kudo and Richardson, 2018) for subword tokenization. The model was finetuned on Dutch sentences as input with the English translation as output for 100 epochs with a batch size of 32 and a starting learning rate of $1 \cdot 10^{-4}$. Data augmentation

---

2. https://labelstud.io
3. https://huggingface.co/Helsinki-NLP/opus-mt-nl-en

consisted of the introduction of spelling errors in the input sentence, changing the input and output sentences to upper, lower, or title case, adding parentheses around all words or the full sentence, and using the output sentence also as input. To reduce translation errors with respect to numbers or abbreviations (e.g., age, measured distances, percentages, and names of genes), we additionally included a set of template sentence pairs with a range of possible values inserted during training. The HuggingFace framework was used for model training and inference. (Wolf et al., 2020)

### 2.3 Segmentation Model

For segmenting subsentences in the translated pathology reports based on the information content, we built upon `FLAN-t5-large`[4] (Chung et al., 2024), which is an instruction-finetuned Transformer model (Vaswani et al., 2017) with an encoder-decoder architecture of 783 million parameters, pre-trained on numerous language tasks. Custom tokens were added to the default SentencePiece tokenizer (Kudo and Richardson, 2018), primarily for the class labels. The model was finetuned on input-output pairs to learn the pathology report-specific subsentence segmentation task for 10 epochs with a batch size of 12 and a learning rate of $1 \cdot 10^{-4}$ at the start. Input prompts consisted of a single sentence to be segmented, in combination with a context window of sentences that precede and follow it. The output format includes the single sentence again, where now each subsentence segment is concluded by the corresponding class token and delimited by a newline token except at the end. An example input-output pair with a context window size of 1 can be seen below:

Input: `"segment_pathology_sentence:'A cutaneous metastasis cannot be excluded for certain, but there is no history of prior melanoma.' Context:'The Breslow thickness is at least 1.6 mm. <SENTENCE> No ulceration, no regression, no satellites.'"`

Output: `"A cutaneous metastasis cannot be excluded for certain,<CON><Nline>but there is no history of prior melanoma.<HIS>"`

Based on a hyperparameter search, evaluated on the validation set, we selected a context window of 7 (from a range of 0-10) and oversampled sentences with annotations of medical history (HIS) and/or calculations and counts (CAL) (i.e., the two smallest classes) by a factor of 5 during training. The `FLAN-t5-large` variant of the model showed slightly better performance than smaller variants. In addition, we investigated including the section name in the input prompt, which improved the performance when using a small context window size, but did not show benefit for larger context windows. The HuggingFace framework was used for model training and inference. (Wolf et al., 2020)

## 3 Results

### 3.1 Translation Performance

The predictive performance of the translation model was evaluated on the independent test set of 3,597 sentences from 150 reports by counting and categorizing the type of errors.

---

4. https://huggingface.co/google/flan-t5-large

Table 2: Segmentation model performance on the test set.

| Class | Words | Precision (95% CI) | | Recall (95% CI) | | F$_1$-score (95% CI) | |
|-------|-------|-----------|----------------|--------|---------------|--------|----------------|
| H&E | 16,575 | 0.976 | (0.965-0.984) | 0.959 | (0.946-0.971) | 0.967 | (0.959-0.975) |
| IHC | 4,959 | 0.958 | (0.939-0.975) | 0.968 | (0.955-0.980) | 0.963 | (0.951-0.974) |
| IHC+ | 977 | 0.732 | (0.636-0.834) | 0.854 | (0.768-0.929) | 0.788 | (0.719-0.855) |
| MOL | 1,132 | 0.929 | (0.870-0.976) | 0.947 | (0.915-0.976) | 0.938 | (0.901-0.966) |
| CLN | 1,515 | 0.987 | (0.973-0.999) | 0.962 | (0.936-0.983) | 0.974 | (0.960-0.986) |
| HIS | 84 | 0.879 | (0.529-1.000) | 0.690 | (0.283-1.000) | 0.773 | (0.400-0.976) |
| CON | 5,326 | 0.922 | (0.895-0.946) | 0.934 | (0.904-0.959) | 0.928 | (0.908-0.946) |
| RAD | 3,387 | 0.965 | (0.947-0.983) | 0.965 | (0.946-0.981) | 0.965 | (0.949-0.979) |
| BRS | 913 | 0.981 | (0.958-0.999) | 0.955 | (0.915-0.989) | 0.968 | (0.945-0.987) |
| CAL | 164 | 0.968 | (0.935-1.000) | 0.921 | (0.786-0.980) | 0.944 | (0.864-0.972) |
| ADV | 854 | 0.935 | (0.883-0.976) | 0.956 | (0.864-1.000) | 0.945 | (0.885-0.985) |
| SID | 691 | 0.924 | (0.884-0.960) | 0.918 | (0.879-0.953) | 0.921 | (0.894-0.945) |
| UNR | 888 | 0.882 | (0.809-0.948) | 0.927 | (0.888-0.960) | 0.904 | (0.860-0.941) |
| Macro avg. | | 0.926 | (0.896-0.943) | 0.920 | (0.884-0.948) | 0.921 | (0.890-0.940) |
| Weighted avg. | | 0.953 | (0.946-0.961) | 0.952 | (0.944-0.960) | 0.952 | (0.944-0.960) |

Translation errors that changed the meaning of a sentence and were undetectable without knowing the input sentence were counted 21 times (0.6%), which mostly concerned (decimal) numbers, anatomical locations, and missed or introduced negation. Spelling and grammar errors were counted 69 times (1.9%). Incorrect translations that were detectable without knowing the input sentence or missing words that did not considerably change the meaning of the sentence were counted 129 times (3.6%).

## 3.2 Segmentation Performance

The predictive performance of the segmentation model was evaluated on the independent test set at word level using the precision, recall, and F$_1$-score for each class. The overall performance was calculated by macro averaging (i.e., using equal class weights) and weighted averaging (i.e., using proportional class weights) the class scores. Bootstrapping (R = 10,000 samples) was used to calculate 95% confidence intervals (CIs). Out of the 4,660 subsentences in the test set, the model incorrectly reproduced the original subsentence 30 times (0.6%) in the output. These reproduction errors, which did not affect the predicted class tokens, were manually corrected before the evaluation.

The results of the segmentation model evaluation are shown in Table 2. The F$_1$-scores per class ranged between 0.773-0.974, with a macro average F$_1$-score of 0.921 (95% CI, 0.890-0.940) and a weighted average F$_1$-score of 0.952 (95% CI, 0.944-0.960). The model performance was substantially lower for the HIS and IHC+ classes in comparison to the other classes. For words annotated but not predicted as part of the HIS-class, CON and IHC were the most frequent alternatively predicted classes. For words annotated but not predicted as part of the IHC+-class, the most frequently predicted alternative classes were H&E and IHC.

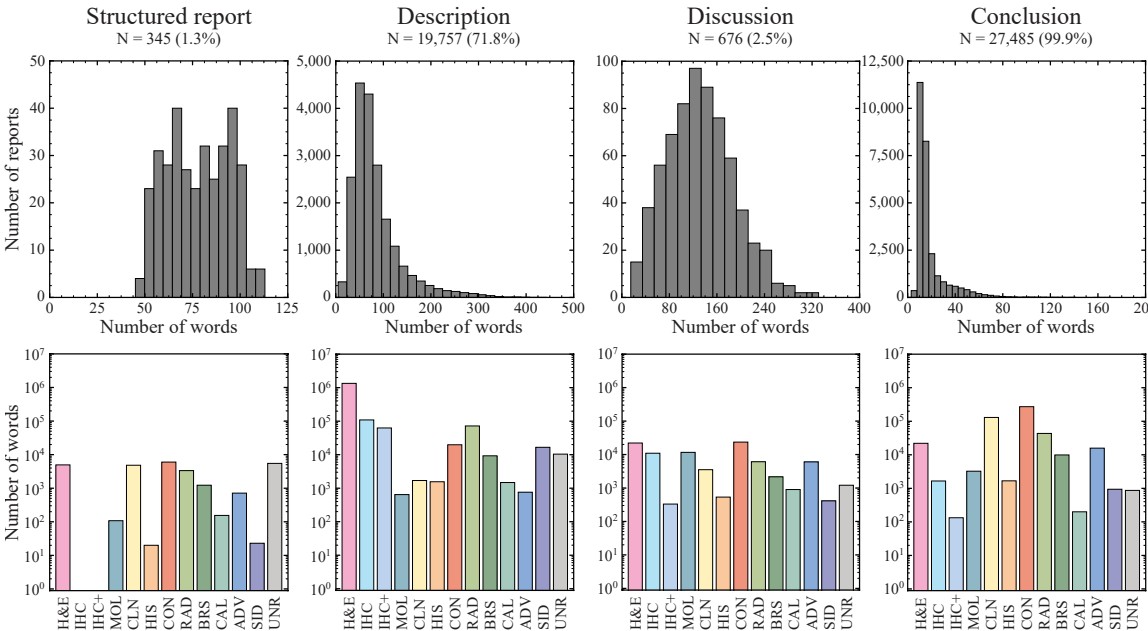

Figure 3: Dataset statistics for the 27,500 melanocytic lesion pathology reports, including histograms with the number of words per report (top row) and bar charts with the number of words per class (bottom row), separated by section type (columns). N indicates the number of reports for which the corresponding section was not empty. Annotations were used for class-specific word counts if available. Model predictions were used for the remaining reports without available annotations. A logarithmic scale was used for the number of words per class.

## 3.3 Dataset Statistics

Statistics for the complete dataset of 27,500 pathology reports for downstream vision-language modeling, extracted after preprocessing of the unannotated reports, are shown in Figure 3. Substantial differences are seen between the sections in terms of the presence, the words per section, and words per class. Almost all reports have a conclusion section (99.9%), which tends to be shorts with a median of 13 words and mostly related to conclusions, clinical information, and the radicality. The majority of the reports also have a description section (71.8%) with a median of 70 words. The descriptions mainly contain observations based on H&E-stained slides, followed by information related to IHC expression patterns and the radicality. The discussion section is much less common (2.5%) and has the largest median length of 128 words. Results of molecular testing and treatment advice are more frequently included in the discussion section compared to the description section. Structured report sections, typically only used for melanomas, are the least common (1.3%) and have a median length of 78 words. In contrast to the other sections, the distribution of the word count for the structured reports does not have a single, clear mode. The class distribution is fairly mixed, with H&E-stained slide observations, clinical information, conclusions, and unrelated information contributing close to equally as largest classes.

IHC-stained slides are not mentioned in the structured reports. Subsentences related to patient history, as well as calculations and counts, are uncommon across all sections.

## 4 Discussion and Conclusion

As part of a preprocessing workflow for pathology reports of melanocytic lesions, we developed separate language models for translation from Dutch to English and subsentence segmentation based on information content. This enables selective use of text data from the reports for vision-language modeling, which can prevent hallucinations in tasks like report generation by training only on information that can be derived from the paired images.

After initial preprocessing, the reports were first translated from Dutch to English using the language model for translation. Before finetuning this model on text from the melanocytic lesion reports, many translation errors were seen with respect to pathology-specific words. The number of errors greatly reduced during the iterative model development as the dataset for finetuning increased in size. In addition, the model was found to be more robust to spelling errors in the input sentences when these errors were also introduced during finetuning as a form of data augmentation. Despite the improvements, words and (decimal) numbers in the input sentences that were not seen during training remained as a source of error and were corrected before segmentation. Hence, we expect the generalizability of the current translation model to other pathology domains to be limited.

The language model for subsentence segmentation showed a strong predictive performance on the independent test set with a macro average $F_1$-score of 0.921 (95% CI, 0.890-0.940) and a weighted average $F_1$-score of 0.952 (95% CI, 0.944-0.960). The model reached a $F_1$-score of above 0.90 for 11 out of the 13 classes, with only a lower performance for the HIS and IHC+ classes. Patient history is only present in a relatively small subset of reports and can be fairly diverse, likely affecting both the model performance and representativeness of the evaluation for the HIS-class. The IHC+-class concerns subsentences with observations based on IHC-stained slides, which are also visible in the H&E-stained slides. Ambiguity regarding some subsentences and possibly also the strong context-dependence could explain the lower segmentation performance for the IHC+-class. A limitation of the presented approach for text segmentation is the possibility of incorrect reproduction of the original sentence in the input. Although this occurred rarely, corrections to the model output were required. A benefit of segmentation for structured reports, which are already intended to ease information retrieval, is robustness to changes in the report format over time.

The developed language models were used to translate and segment the unannotated reports to complete the preprocessing of the entire dataset of 27,500 pathology reports. In future work, we plan to use the preprocessed dataset in combination with the corresponding whole slide images to investigate vision-language modeling for tasks like multimodal pretraining and report generation in the domain of cutaneous melanocytic lesions.

## Acknowledgments and Disclosure of Funding

This research was financially supported by the Hanarth Foundation.

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

## Supplementary Material

Table 3: Number of reports and selection strategy used per iteration in the iterative development approach for the language models.

| Iteration | Selection | Reports | Cumulative Reports |
|---|---|---|---|
| Initial | Random from long reports | 30 | 30 |
| 1 | Random from all reports | 100 | 130 |
| 2 | Random from all reports | 225 | 355 |
| 3 | Random from all reports accessioned in 2020 | 250 | 605 |
| 4 | Random from all reports accessioned in 2020 | 250 | 855 |
| 5 | Random from reports with coincidental findings | 79 | 934 |
| 6 | Random from reports with molecular testing | 194 | 1,128 |
| 7 | Reports with most new Dutch words | 122 | 1,250 |
| 8 | Reports with most new Dutch words | 250 | 1,500 |

