# OpenReview forum: "Preprocessing Pathology Reports for Vision-Language Model Development"
_MICCAI.org/2024/Workshop/COMPAYL — COMPAYL 2024_

### Official Review · Reviewer_sRDN · 2024-07-07
**In this paper, the authors focus on the pre-processing of pathology reports to be used for the development of a vision language model.**

**Custom Rating:** 5
**Confidence:** 5

**Review:**

The study provides a workflow for the pre-processing of pathology reports for the development of a visual language model. The pathology reports were first translated from Dutch into English and then the sub-sentences were segmented. Both tasks were performed by two language models. Both tasks were performed by two language models iteratively over a period of eight iterations. The translation model was scored based on the number of errors of varying severity, while the sub-sentence segmentation model was scored based on F1 scores.

During the review, the following points were raised, and it is recommended to improve the manuscript accordingly:

1.	The authors mentioned that the models were developed iteratively to correct predictions. It would be beneficial to include performance data for each iteration to assess whether the model improved continuously or if improvements were random. Additionally, it would be helpful if the authors clarified what they mean by an iteration, how many reports (from the 1350 annotated reports) were considered for each iteration, and whether this number remained constant throughout all iterations.

2.	Were the reports translated to simplify the development process (if the AI researcher does not know Dutch) or for another reason? What was the main reason for translating the reports from Dutch to English? Are there any development limitations? To understand the entire process, it is important to know whether the model will require an additional model to translate the outcomes back into the original language in practical use.

3.	What were the common errors corrected in the translation when the model was applied to the unannotated 25,850 reports before using the segmentation model? Did these errors affect the performance of the segmentation model?

4.	The HIS class segmentation shows worse performance, likely due to an insufficient number of evaluated samples. Did the authors consider using weighted augmentation or a class weight approach in training the model? Similarly, was the same consideration given for the IHC+ class?

---

### Official Review · Reviewer_XZVy · 2024-07-09
**Iterative preprocessing techniques to improve vision-language models for pathology reports**

**Custom Rating:** 4
**Confidence:** 4

**Review:**

This paper presents a methodology for preprocessing pathology reports to develop vision-language models. The authors focus on subsentence segmentation and translation from Dutch to English for 27,500 pathology reports of cutaneous melanocytic lesions. The preprocessing workflow includes de-identification, translation, and segmentation using iteratively improved language models. The final models achieved high predictive performance, with macro and weighted average F1-scores of 0.921 and 0.952, respectively. The methodology is designed to prevent hallucinations in report generation by ensuring only image-derivable information is used. The approach is generalizable to other pathology domains, and the code and trained models are publicly available.

Strengths
The paper outlines a detailed and systematic approach to preprocessing pathology reports, which is crucial for accurate vision-language model development.
The use of iterative development and manual correction to enhance model performance is a robust approach, leading to significant improvements in translation and segmentation accuracy though it may be exhaustive and resource consuming.
The models achieved good F1-scores, indicating strong predictive capabilities.
Making the code and trained models publicly available promotes transparency and facilitates further research in the field.
The methodology is designed to be applicable to various pathology domains, not just melanocytic lesions, enhancing its utility.

Weaknesses (Constructive Feedback)
The need for manual correction of model outputs indicates a dependency on human intervention. Though it is a trustful way however, developing more robust error detection and correction mechanisms within the models could reduce this dependency.
While the dataset is extensive, it is specific to a single institution and pathology type. Expanding the dataset to include reports from multiple institutions and diverse pathology types could enhance the robustness and applicability of the models.
The paper primarily lies in data collection and preparation rather than the development of new models. In addition to the mentioned future work of combining WSIs for multimodal prediction, authors could focus on innovating the model architecture or introducing novel techniques in the preprocessing steps.
The translation model’s generalizability to other pathology domains is limited. Exploring domain adaptation techniques or multi-domain training for enhanced generalizability can be additional contributions.
As the segmentation model showed lower performance for certain classes (HIS and IHC+), further investigation on the reasons can be beneficial for the readers. Moreover, incorporating additional training data or context-specific adjustments could be ways to think of for improving the performance.

Final thoughts:
This paper seems to be an intermediate report for a larger direction. The big picture of the project seems interesting. Although this paper is limited to the preprocessing and clean data collection techniques, as the details are well described and the code will be released it can benefit the community. The release of the whole dataset will also be beneficial.

---

### Official Review · Reviewer_P2RE · 2024-07-09
**Review of paper 9**

**Custom Rating:** 5
**Confidence:** 3

**Review:**

The article addresses the issues of Dutch-to-English translation of pathology reports and the segmentation of translated reports by fine-tuning large language models, and conducts a detailed evaluation of their performance. The project is substantial and holds promising application potential. Importantly, both models are publicly accessible, providing valuable resources and baselines for the integration of pathology and NLP.

I recommend accepting the paper, but some details should be expressed more clearly before publication.

1. The process of manual correction is unclear. During each iteration, are the mistakenly predicted samples identified from the unannotated set and then added to the training and validation sets? If that is the case, I would like to see a figure showing the increasing performance during iterations between model and pathologists (with more corrected developing set samples).

2. The article mentions that the development set ultimately comprises 1500 reports, consistent with Table 1. Starting with how many samples did the iterative correction process begin? How many samples were added to the training and validation sets in each iteration? (whether errors happens frequently? if so, which sample have priority to be corrected?)

3. Section 3.1 mentions translation errors. For translation tasks with non-unique solutions, how are errors defined and evaluated?

Additionally, I would like to inquire whether the datasets involved in the study could potentially be made publicly available (in future publications). This would be a highly valuable resource for the community.

---

### Decision · Program_Chairs · 2024-07-16

Accept